# Systematic Guidelines for Effective Utilization of COVID-19 Databases in Genomic, Epidemiologic, and Clinical Research

**DOI:** 10.3390/v15030692

**Published:** 2023-03-06

**Authors:** Do Young Seong, Jongkeun Park, Kijong Yi, Dongwan Hong

**Affiliations:** 1Department of Medical Informatics, College of Medicine, Catholic University of Korea, 222 Banpo-daero, Seocho-gu, Seoul 06591, Republic of Korea; 2Graduate School of Medical Science and Engineering, Korea Advanced Institute and Technology (KAIST), Daejeon 34141, Republic of Korea; 3Precision Medicine Research Center, College of Medicine, Catholic University of Korea, 222 Banpo-daero, Seocho-gu, Seoul 06591, Republic of Korea; 4Cancer Evolution Research Center, College of Medicine, Catholic University of Korea, 222 Banpo-daero, Seocho-gu, Seoul 06591, Republic of Korea

**Keywords:** SARS-CoV-2, COVID-19 database, data categorization, function utilization, multiple queries, integrative analysis method

## Abstract

The pandemic has led to the production and accumulation of various types of data related to coronavirus disease 2019 (COVID-19). To understand the features and characteristics of COVID-19 data, we summarized representative databases and determined the data types, purpose, and utilization details of each database. In addition, we categorized COVID-19 associated databases into epidemiological data, genome and protein data, and drug and target data. We found that the data present in each of these databases have nine separate purposes (clade/variant/lineage, genome browser, protein structure, epidemiological data, visualization, data analysis tool, treatment, literature, and immunity) according to the types of data. Utilizing the databases we investigated, we created four queries as integrative analysis methods that aimed to answer important scientific questions related to COVID-19. Our queries can make effective use of multiple databases to produce valuable results that can reveal novel findings through comprehensive analysis. This allows clinical researchers, epidemiologists, and clinicians to have easy access to COVID-19 data without requiring expert knowledge in computing or data science. We expect that users will be able to reference our examples to construct their own integrative analysis methods, which will act as a basis for further scientific inquiry and data searching.

## 1. Introduction

In 2019, with the spread of the coronavirus disease 2019 (COVID-19) pandemic, it became crucial for the scientific community to quickly have access to accurate, detailed COVID-19 data [1,2]. Access to epidemiological data proved to be important for tracking the pandemic (rates of infection and the pandemic response of different hotspots/countries) [3,4]. The collection of high-quality genomic data generated a better understanding of severe acute respiratory syndrome coronavirus 2 (SARS-CoV-2) that causes COVID-19 [5,6]. Protein structure and sequence data improved the understanding of SARS-CoV-2 and its components [7,8]. Clinical trials and drug data helped clinicians and researchers develop treatments during the pandemic [9,10].

Our understanding of COVID-19 developed over a short period of time, making early response to the pandemic challenging. The mRNA vaccines used worldwide were utilized without fully going through testing [11]. It has also been challenging to accurately assess the effectiveness of the vaccines and other treatments for COVID-19 [12]. Through obtaining epidemiological, genomic, and clinical data on COVID-19, managing the pandemic has become possible. However, it is not evidently clear how effective accumulation of COVID-19 data can be, so a comprehensive understanding of the data is required.

The abundance of various types of COVID-19 data has led to an increased number of databases. An increase in genomic data submission has resulted in databases with large amounts of data, a multitude of filters and search options, and numerous downloadable metadata or associated files [13]. This can potentially be challenging for users to access the data they need. The rate at which COVID-19-related publications are accepted and published by far outpaces that of previous outbreaks, such as the Ebola or Zika viruses [14,15]. This has led to an increased need for the organization of the literature. We investigated the amount of genomic data being collected by the National Center for Biotechnology Information (NCBI), as well as virus research and the number of COVID-19 publications submitted on a monthly basis (Appendix A). We also compiled the amount of epidemiological data, genome and protein data, treatment data, and publications that have been submitted to various COVID-19 databases (Table 1).

## 2. A Survey of COVID-19 Databases

Databases and research organizations that existed before the COVID-19 pandemic played a part in the initial data collection process. The Global Initiative on Sharing All Influenza Data (GISAID), which is responsible for the curation of genomic data on previous influenza outbreaks and virus research, possessed the necessary infrastructure to start collecting data on SARS-CoV-2 sequences [16,17].

Organizations, such as the National Institutes of Health (NIH) and European Molecular Biology Laboratory (EMBL) were poised to be hubs for tools and resources in response to COVID-19 [18]. Global organizations, such as the World Health Organization (WHO), which we usually look to for guidance and resources in worldwide health issues, also played a part. During previous epidemics, organizations, such as the WHO gathered data on cases and deaths, improving the understanding of the pathogen’s infectivity and lethality. Information on vaccine development and vaccine doses helped governments keep track of vaccination progress within populations [19,20].

There has been a rapid emergence of COVID-19-related databases with the fast accumulation of such data. Genomic sequences are being submitted to GISAID faster compared to other viruses. Raw sequence data on SARS-CoV-2 and its variants have been submitted to GISAID, Nextstrain, the University of California Santa Cruz (UCSC) Genome Browser for COVID-19 Research on the UCSC page, and the COVID-19 Data Portal powered by EMBL-European Bioinformatics Institute (EBI).

Databases, such as LitCovid, developed by the NIH, were created to exclusively organize and help users search for COVID-19 publications [21,22]. SARS-CoV-2 has a wide range of variants with numerous classifications of variants of concern (VOCs), such as Alpha, Beta, Gamma, Delta, and Omicron, and variants of interest (VOIs) [23,24,25]. There are various organizations and databases that have different nomenclature for classifying variants. The WHO, Pangolin, and Nextclade, powered by Nextstrain, are considered to be the main classifying entities [26,27,28]. Epidemiological data have been collected by various governments and institutes and then collected and centralized in databases, such as those of the WHO. The gathering of COVID-19 case data enables a better understanding of the infectivity of SARS-CoV-2 and its variants [29]. COVID-19 death data likewise help keep track of the lethality of the variants. With the numerous types of vaccines and the need for multiple vaccinations, vaccine data on COVID-19 have been instrumental in keeping track of global vaccination progress [30,31].

With COVID-19 databases, there has been a transition from text-based visualization to more dynamic visualizations with charts and graphs. Alongside more user-friendly interfaces, there has been improvement in visualization tools, internal database self-developed tools, and more effective analysis tools. Nextstrain visualizes its genomic data as phylogenetic trees that display clade and variant information. Users viewing the phylogenetic tree can interact with it to change display options, time, and the variants being displayed to visualize what they need.

We decided that detailed information of the contents of each of the 15 databases we investigated would be too extensive to include in the main article. We have included detailed explanations of the COVID-19 database features alongside screenshots of the appropriate web pages (Appendix A).

## 3. COVID-19 Database Categorization and Main Purposes

We selected 15 databases that are most frequently used out of all of the currently available COVID-19 databases. Based on our investigations of the databases’ main purposes and tools, we categorized the databases on the basis of their data characteristics and then reported their specific functions, internally developed tools, and associated websites.

We investigated the main COVID-19 databases and then divided them into three categories: epidemiological data, genome and protein data, and drug and target data. There were databases that fit only into one category and those that displayed more than one characteristic. For each category, we provided explanations on what type of data was included and sorted. First, we categorized cases, deaths, hospitalizations, vaccinations, vaccine dose data, etc., on a country or time basis as epidemiological data. We determined the databases containing these data to be the Johns Hopkins Coronavirus Resource Center, Our World in Data, GISAID, WHO, and the National Center for Biotechnology Information (NCBI) SARS-CoV-2 Resources. Second, we categorized genome and protein data as data on SARS-CoV-2 amino acid sequences, protein domains and regions, and mutations in SARS-CoV-2 lineages and clades. We determined that GISAID, the WHO, the NCBI SARS-CoV-2 Resources, interaction data between coronavirus RNAs and host proteins (CovInter), T-cell COVID-19 Atlas (T-CoV), Immune escape variants in SARS-CoV-2 (ESC), DockCoV2, USCS Genome Browser, COVID-19 Data Portal, cov-lineages.org, Protein Data Bank (PDB), COVID CG, and Nextstrain included this category of data. Third, we categorized drug and target data to include data on antibodies, drugs, drug trials, drug-protein interactions, target molecules, etc., and CovInter, T-CoV, ESC, DockCoV2, the WHO, and NCBI SARS-CoV-2 Resources included drug and target data (Figure 1a, Table 2 and Appendix A).

COVID-19 databases can be sorted into databases that make use of data provided by GISAID and databases that independently collect their own data. We investigated the purposes of the data present in these databases and classified them into nine distinct groups: (1) clade/variant/lineage, (2) genome browser (sequence), (3) protein structure, (4) epidemiological data (cases, vaccine rates, deaths), (5) visualization, (6) data analysis tool, (7) treatment (clinical trials, drugs), (8) literature, and (9) immunity (Figure 1b,c). In addition to the specific purposes we compiled and categorized for the databases, many of the databases possess software, tools, and functions that aid in data viewing and analysis (Appendix A).

In our categorization of the 15 databases, we noted that there was a heavy focus on genome and protein data. GISAID, while considered to possess one of the most extensive datasets on SARS-CoV-2 genome sequences, is not easily accessible. The data cannot be accessed without creating an account first, and there are limitations with regard to downloading the data. Meanwhile, many of the databases source their data from GISAID or perform analysis using sequence data provided by GISAID (Figure 1b). With epidemiological data, the Johns Hopkins University Coronavirus Resource Center focuses heavily on U.S. datasets. With Our World in Data and the WHO database, there are slight discrepancies between datasets and limitations on data collection. For drug and target data, there is no a centralized database that has properly compiled data on COVID-19 preventative measures. Of the databases we investigated, databases, such as CovInter are not easily found simply through search engines.

Many, if not most of the COVID-19 databases, are specialized and built for specific types of data. Data collection and analysis are performed in line with the purpose of the database. Depending on who is accessing the data, there is a difference in what type of data are needed and prioritized for the user. Researchers, epidemiologists, and clinicians each have the respective data they need [50]. Because of these issues, it is necessary to have a centralized system that can make comprehensive use of the available databases. We believe that it is important to provide guidance on how to navigate between and utilize databases. Specifically, we introduced different databases, how to use them, and where they can be used.

We aimed to introduce the currently available COVID-19 databases and categorize them according to their purposes. By introducing their main functions and tools, users can become more familiar with the wide range of databases that exist. We showed how to utilize the databases through specifically built queries and examples, dependent on the purpose of the user. The queries and examples we showed require no expert knowledge or computing skills from the users to utilize the databases.

## 4. Advanced Utilization of COVID-19 Databases

We aimed to help users with their database utilization by providing specific examples of utilizing the databases we have introduced. In addition to the examples we provided, it is important for users to learn personalized database utilization. We expect that clinical researchers, epidemiologists, and clinicians will find specific uses for our examples [51,52]. Not only will users be able to understand the specific queries we presented but they will also be able to devise their own queries and fully utilize the databases we have introduced. In addition to the four queries we showed, we built additional queries (Appendix A).

“Periods of high numbers of patient deaths and severe cases during the COVID-19 pandemic can be used as a reference to investigate lineage/clade variants to identify dangerous variants and related candidate variants.”

During the COVID-19 pandemic, patient deaths and severe cases occurred in several waves [53]. Worldwide, vaccination began in 2021, with vaccines continuing to be administered at present [54,55,56]. Nevertheless, COVID-19 patient numbers are still on the rise. Numerous COVID-19 variants have emerged through genetic evolution, and in that process, new lineages have disappeared and appeared. Utilizing the WHO database, current and past variants of concerns (VOCs) can be identified [57]. By examining these variants, lineages during periods of high rates of deaths and severe COVID-19 cases can be determined. Research identifying candidates for variants with high associated COVID-19 risks is necessary [58,59]. Therefore, we carried out the following analysis integrating various databases, which will help with the optimization of dealing with future variants and help antibody and drug development [60,61,62].

(1) To identify VOCs, we accessed the WHO Tracking SARS-CoV-2 Variants page to check Alpha (B.1.1.7, 8 December 2020), Beta (B.1.351, 18 December 2020), Gamma (P.1, 11 January 2021), Delta (B.1.617.2, 11 May 2021), and Omicron (B.1.1.519, 26 November 2021) variants and check the lineages and VOC dates;

(2) To identify the mutations in VOCs, we navigated to cov-lineages.org to identify characteristic mutations for the Alpha, Beta, Gamma, Delta, and Omicron lineages;

(3) To obtain data on VOCs, we checked Nextstrain’s latest global analysis table. In the upper left corner, by selecting the dataset (ncov, open or GISAID, global or country), date range (22 December 2019 to 7 November 2022), and color (GISAID, Pango lineage, etc.), we could identify the first reported date and the latest date for VOCs through phylogeny;

(4) We checked epidemiological data by navigating to the Our World in Data database’s COVID-19 Data Explorer Table. We used COVID-19 Data Explorer to check the number of deaths, sorted by international or country, and selected confirmed deaths for metrics. The date range provides data from 28 January 2020 to 11 November 2022;

(5) Upon investigating Pango lineage by deaths per 1 million people from 28 January 2020 to 11 November 2022, we verified that the Delta (21A), Delta (21J), Omicron (21M), Omicron (21K), Omicron (21L), and Omicron (22B) lineages fit in the time period. We could predict our candidate variants by analyzing the lineages that fit in the time period. Upon analyzing the lineages during the periods of high numbers of confirmed deaths, for the spike protein area, the common variants were T478K and D614G. The variants that had different amino acid substitutions in the same location were P681R (Delta) and P681H (Omicron). For the N protein, R203M was found (Figure 2).

① The WHO Tracking SARS-CoV-2 Variants page was utilized to check currently circulating variants of concern (VOCs) and previously circulating VOCs (the currently circulating VOC is Omicron);

② To check the mutations of Omicron’s sublineages (BA.1, BA.2, BA.3, BA.4, and BA.5), cov-lineages.org was used. Upon clicking on cov-lineages.org Pango, the characteristic mutation table data were utilized to search for each lineage of BA.1, BA.2, BA.3, BA.4, and BA.5;

③ To check Omicron’s sublineage data and date information, Nextstrain was used. Upon clicking on Nextstrain’s latest global analysis—GISAID or latest global analysis—open data, the SARS-CoV-2 genomic epidemiology was checked. The data can be filtered by dataset, data range, color by options, etc.;

④ To obtain data on global deaths from the time period when Omicron was dominant, Our World in Data was used. By clicking on the COVID-19 Data Explorer tab, the COVID-19 data explorer is accessed. In options, country name was selected for sorting, confirmed deaths were selected for metrics, and the 7-day rolling average was selected for intervals;

⑤ Based on these results, comprehensive analysis was possible. Through the overlap of lineages’ date of origin and period of time for COVID-19 deaths, common variants were extracted. A route for finding dangerous variants was provided.

This query will be the most useful for targets in the descending order of clinical researchers, epidemiologists, and clinicians.

2.“For representative SARS-CoV-2 Delta and Omicron lineages, we checked virus and host protein interactions, and after investigating the related publications, we explained the relevance to SARS-CoV-2.”

A well-known SARS-CoV-2 infection pathway involves the coronavirus spike protein detecting the angiotensin-converting enzyme 2 (ACE-2) receptor in the host membrane and entering the host cell to cause infection [63,64,65,66,67]. Numerous mutations have occurred in SARS-CoV-2 through evolution since its outbreak [68,69,70]. These mutations affect 3-D protein structures and can cause problems: vaccine and treatment effectiveness can be weakened, leading to an increase in the number of infections [71,72,73]. From this perspective, an understanding of the virus pathogen structure, infection process and pathway, and interaction is needed [74,75,76]. To investigate viral molecular cell biology, we used various databases to check virus and host interactions. We can then extract the candidate protein for utilization in the development of new treatments [77,78].

(1) We utilized cov-lineages.org to identify the Pango lineages of the WHO nomenclature Delta and Omicron variants to be Delta B.1.617.2 and Omicron BA.1, BA.2, BA.3, BA.4, and BA.5;

(2) We utilized CovInter to investigate the interaction between coronavirus and the host protein in the spike protein, where the highest number of SARS-CoV-2 mutations are found. We selected the Delta (B.1.617.2 and AY lineages) strain hCoV-19/Namibia/N17380/2021 and Omicron (B.1.1.529; BA.1; BA.1.1; BA.2; BA.3; BA.4; BA.5) strain hCoV-19/Zimbabwe/CERI-KRISP-K034087/2021;

(3) From the dates of Delta hCoV-19/Namibia/N17380/2021 and Omicron hCoV-19/Zimbabwe/CERI-KRISP-K034087/2021, we can locate and analyze the virus RNA host protein network;

(4) Using the LitCovid database, we searched for the host IMP-1 protein that interacts with both the Delta and Omicron variants, specifically for publications with wet lab experiments;

(5) Through analysis of the Delta and Omicron variants, candidate proteins that interact with both lineages IMP-1, SND1, EIF3D, EIF3H, FASTKD2, GRWD1, IMP-2, G15, LSM11, and UTP18 were found. By extracting data on these interacting proteins, we can use them to supply data for vaccine and treatment development that is not affected by the emergence of variants (Figure 3).

① To check the mutations present in the Omicron variant, cov-lineages.org was used. Upon clicking on Pango, Delta and Omicron were searched for in the lineage list. For each lineage, data from the characteristic mutation table were obtained;

② To examine the interaction between coronavirus RNA and the host protein, CovInter was used. To check the spike protein region, Search Virus RNA was selected, and the S region was clicked on. Among those, Delta and Omicron analysis samples were clicked on to extract the host protein list from each;

③ To check candidate proteins’ connection to SARS-CoV-2, publications were searched for through LitCovid;

④ Consequently, through an interaction between coronavirus RNA and host protein and Delta and Omicron comparisons, unique proteins and shared proteins were found. Shared proteins can be utilized in vaccines or treatments against multiple SARS-CoV-2 variants.

This query will be the most useful for targets in the descending order of clinical researchers, clinicians, and epidemiologists.

3.“For the currently circulating VOC Omicron, taking vaccination rates and reproduction rates into consideration, we can obtain effective antibody and drug data based on clinical trial information and Omicron genetic mutation.”

The current dominant SARS-CoV-2 strain worldwide is BA.5 [79,80,81]. Many countries are carrying out vaccinations as a preventive measure against COVID-19 with vaccine and treatment development also underway [82,83]. Many mutations occur in SARS-CoV-2 through evolution, which makes preexisting vaccines ineffective. To combat the continued evolution of SARS-CoV-2, research that targets the mutations is needed [84]. To select effective antibodies and drugs based on mutation data on reported lineages, we suggest the following process [85,86,87].

(1) Through the WHO’s Tracking SARS-CoV-2 Variants page, we can verify that Omicron is the currently circulating VOC;

(2) To obtain the latest data on the Omicron lineage, we searched for BA.5 information through Nextstrain, specifically for BA.5 date information;

(3) From the BA.5 date of origin, we checked vaccination and reproduction rates in the Our World in Data database. BA.5 is indicated in the figure with the vaccine dose in yellow and the reproduction rate in green. Worldwide, new vaccine dose rates have declined in 2022, while the reproduction rate has been consistent thus far;

(4) Through Nextstrain, we checked not just BA.5 but also BA.2, so we used cov-lineages.org to obtain and analyze data on BA.5 and BA.2 variants;

(5) Upon investigating the mutations of the currently dominant strains BA.5 and BA.2, many shared mutations were found in the spike protein. We utilized the ESC database to search for antibodies and related drugs against T478K among the shared mutations. Entering T478K in the search bar shows related vaccines and antibodies, and for drugs related to the T478K mutation, etesevimab data are provided;.

(6) We can use the NIH’s ClinicalTrials.gov to find a study feature to look for etesevimab and current progress on COVID-19 clinical trials;

(7) For effective measures against COVID-19 variants that occur through mutation, research on cures and prevention must be carried out. Excluding the shared mutations for BA.5 and BA.2, if we investigate the unique mutations in the spike protein, we can find del69/70, L452R, and F486V for BA.5 and Q493R for BA.2. For unique mutations of BA.5, antibodies and treatments, such as bamlanivimab, cilgavimab, tixagevimab, and casirivimab are found, while for unique mutations of BA.2, candidate antibodies or treatments do not exist. Therefore, to maximize the effectiveness of treatments, we need data on each variant (Figure 4).

① Through the WHO Tracking SARS-CoV-2 Variants, the currently circulating VOC (Omicron) was checked;

② From information on the currently circulating VOC Omicron, Nextstrain was used to check BA.5 and BA.2. Upon clicking on Nextstrain’s latest global analysis—GISAID tab or latest global analysis—open data tab, genomic epidemiology data of BA.5 and BA.2 can be checked;

③ Our World in Data was used to check the connection with the period of time of BA.5 and BA.2 prevalence and vaccine dose and the reproduction rate among the epidemiological data;

④ Mutation data on the BA.5 and BA.2 variants were collected from cov-lineages.org;

⑤ From these mutations, mutations found in common were extracted (T478K). Antibody and drug data on the extracted mutations were collected from ESC (etesevimab);

⑥ To determine whether there were clinical trial data on etesevimab, ClinicalTrials.gov was used;

⑦ Regarding vaccine doses, vaccination rates are decreasing globally, whereas there are no significant changes in reproduction rates. Upon investigating the mutations of dominant strains BA.5 and BA.2, antibodies or drugs that target the unique mutations can be searched for. Based on these results, antibodies and drugs that are effective against the variant type or characteristic can be presented.

This query will be the most useful for targets in the descending order of clinicians, clinical researchers, and epidemiologists.

4.“For the VOC Omicron, we obtained data on Pango lineages BA.1, BA.2, BA.3, BA.4, and BA.5, and from current worldwide policy indexes, such as the stringency index and containment and health index and vaccination policy, and we obtained data on countries’ policy responses to COVID-19.”

Following the start of the COVID-19 pandemic, nations worldwide implemented various policies in response to the pandemic [88]. Initial policy responses entailed stricter measures, such as restrictions on gatherings, stay-at-home requirements, and travel restrictions [89,90,91]. Following the vaccine development, with the increase in worldwide vaccinations, policies have changed and become endemic [92,93,94]. However, there were indications that COVID-19 might break out again in late 2022, so constant care and attention are needed. For our example, we planned to investigate countries’ current policy responses based on the currently circulating VOC Omicron and its associated Pango lineages [95,96,97]. We examined the G20 nations in Asia (South Korea, China, Japan, India, Indonesia, Turkey, and Saudi Arabia), Europe ( France, Germany, Italy, the United Kingdom, and Russia), the Americas (the United States, Canada, Mexico, Argentina, and Brazil), Africa (South Africa), and Oceania (Australia).

(1) To investigate the VOCs, we used the WHO Tracking SARS-CoV-2 Variants page to verify that the currently circulating VOC is Omicron, including sublineages BA.1, BA.2, BA.3, BA.4, and BA.5;

(2) Utilizing Nextstrain, we verified the earliest date and most recently observed date for the currently dominant strains BA.1, BA.2, BA.3, BA.4, and BA.5;

(3) To view the worldwide policy response situation, we navigated to Our World in Data’s Coronavirus Pandemic (COVID-19) page and selected the Policy Responses Table. The Policy Responses tab includes information, such as the stringency index, containment and health index, vaccination policy, and Google mobility trends;

(4) Stringency index scores, calculated by the metrics of school closures, workplace closures, cancellation of public events, restrictions on public gatherings, closures of public transport, stay-at-home requirements, public information campaigns, restrictions on internal movements, and international travel controls, are decreasing for G20 countries regardless of vaccination rates. The containment and health index indicates the strictness of government responses, with the strictest country being China. All nations worldwide, excluding a few countries, such as Germany, Egypt, Senegal, and Sierra Leone, currently have universal availability as their vaccination policy. Childhood vaccination eligibility and vaccination eligibility for each country are visually displayed;

(5) Based on the VOC Omicron and the date information of the currently dominant strains BA.2 and BA.5, we investigated the governmental policy responses at the time;

(6) Prior to the emergence of BA.2 and BA.5, deaths from COVID-19 were on the decline in G20 countries, with the trend continuing after their emergence (excluding countries that do not have data). Following the emergence of BA.2 and BA.5, the stringency index and containment and health index became less strict. For infection management, policy guidelines should be established not merely based on COVID-19 deaths, but on the basis of COVID-19 genome and expression data (Figure 5).

① Through the WHO Tracking SARS-CoV-2 Variants page, the currently circulating VOC Omicron can be checked with its lineages BA.1, BA.2, BA.3, BA.4, and BA.5;

② To check the sublineages and date information on the currently circulating VOC Omicron, Nextstrain was used. Upon clicking on Nextstrain’s latest global analysis—GISAID tab or latest global analysis—open data tab, BA.5 and BA.2 data can be accessed;

③ To check data on policies among epidemiological data, Our World in Data was used;

④ By clicking on Our World in Data’s Policy Responses tab, data on numerous COVID-19 policies can be obtained. The stringency index, containment and health index, and vaccination policy were utilized (G20 countries were selected);

⑤ From the stringency index data, after the emergence of BA.5 and BA.2, all countries have become less strict in their policies and containment and health index measures. Since deaths from BA.5 and BA.2 have remained low, relaxed policies related to vaccination or hygiene can provide direction.

This query will be the most useful for targets in the descending order of epidemiologists, clinicians, and clinical researchers.

## 5. Conclusions

Currently, COVID-19 data are being continuously accumulated in worldwide databases. We have drawn comparisons between 15 primary databases, organizing them into categories and investigating the data present. The challenge posed by utilizing these databases is the difficulty in locating and accessing the necessary data due to how they are specialized. We designed queries, guidelines, and Appendix A to address these issues. The integration of multiple COVID-19 databases makes use of the individual strengths of the databases we investigated and categorized to comprehensively present methods to answer important scientific questions. The importance of our integrative analysis of various COVID-19 databases is that clinical researchers, epidemiologists, and clinicians working on COVID-19 have increased accessibility to SARS-CoV-2 and COVID-19 data and the ability to effectively utilize databases.

COVID-19 databases are specialized to present specific datasets, yet there is a need to comprehensively access databases. We present our methodology as an easy, effective access to necessary COVID-19 data without expertise in computing or data science. To prevent the recurrence of the pandemic, it is important for researchers, clinicians, epidemiologists, pharmaceutical companies, and public officers to have fast access to necessary data. For a rapidly mutating virus, such as SARS-CoV-2, it is crucial to keep track of emerging variants and mutation data. Furthermore, it becomes possible to find and recommend necessary treatments utilizing COVID-19 data. The methodology investigated and explained in our study is expected to be used by various researchers and be the basis for building infrastructure and resources to effectively deal with any future epidemics that may occur. A new pandemic will lead to a compilation of epidemiologic data and genomic data, necessitating data categorization and interpretation. The queries and guidelines we have presented for navigating and utilizing databases will therefore be helpful for tracking patients and the disease, finding suitable treatments, and data analysis.

## Figures and Tables

**Figure 1 viruses-15-00692-f001:**
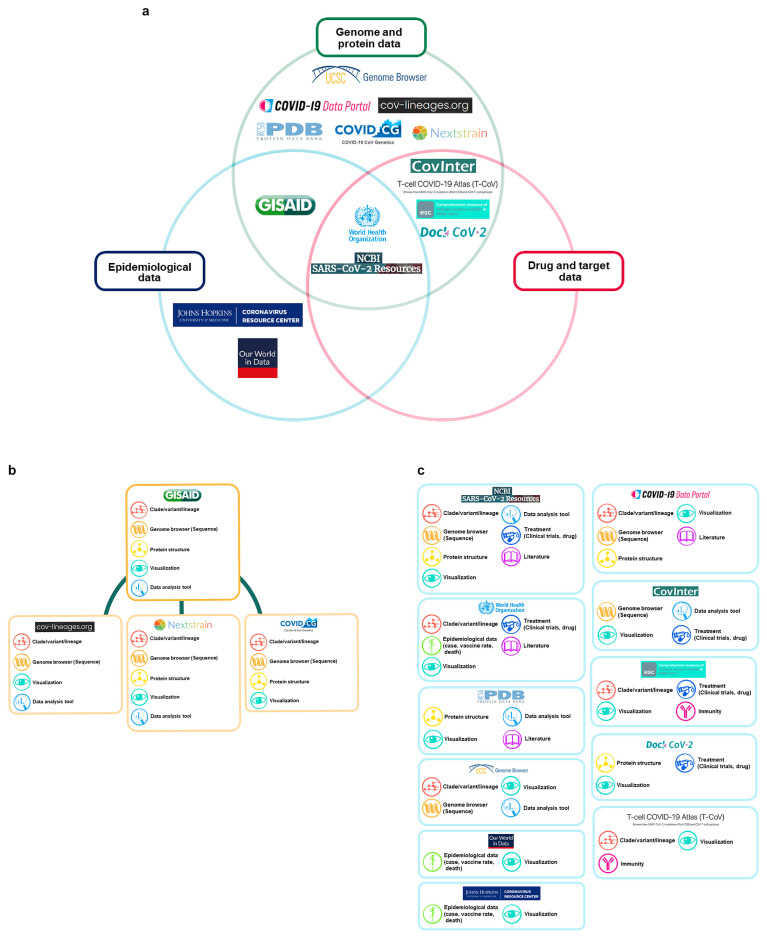
COVID-19 database categorization and main functions. (**a**) Upon investigating 15 databases, these were categorized into three categories: genome and protein data, epidemiological data, and drug and target data. (**b**) There are four databases based on GISAID data. The main functions of the databases are clade/variant/lineage, genome browser (sequence), protein structure, visualization, and data analysis. (**c**) There are 11 databases that have collected their data independently. The main functions are clade/variant/lineage information, genome browser (sequence), protein structure, epidemiological data (cases, vaccine rates, deaths), visualization, data analysis tool, treatment (clinical trials, drugs), literature, and immunity. These main functions vary depending on the purpose of the database.

**Figure 2 viruses-15-00692-f002:**
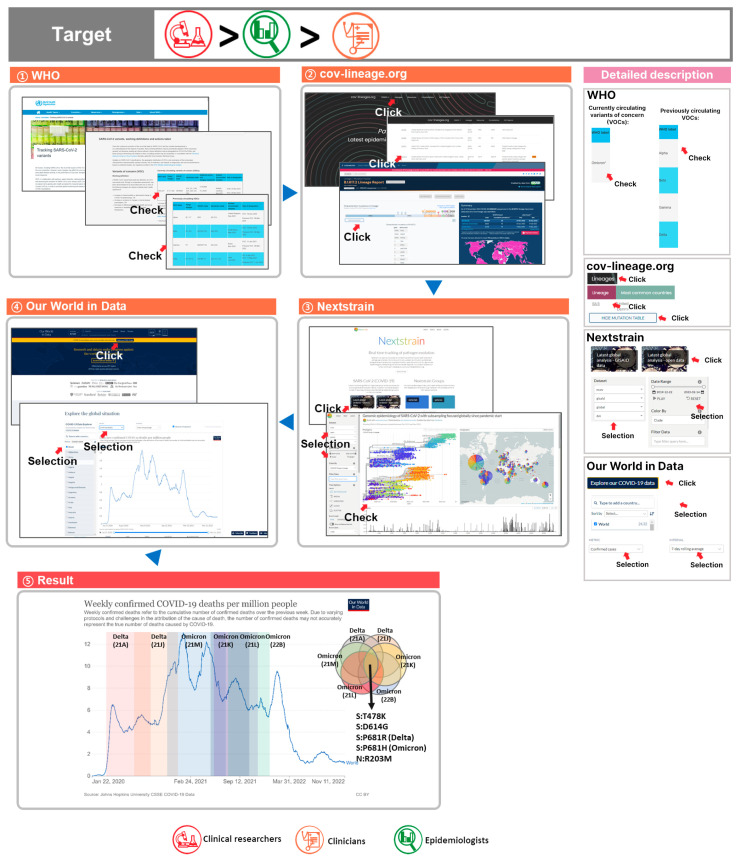
Comprehensive utilization of COVID-19 databases 1. Through an analysis of variants and deaths in certain periods of time, the extraction of important variants was carried out in the following process.

**Figure 3 viruses-15-00692-f003:**
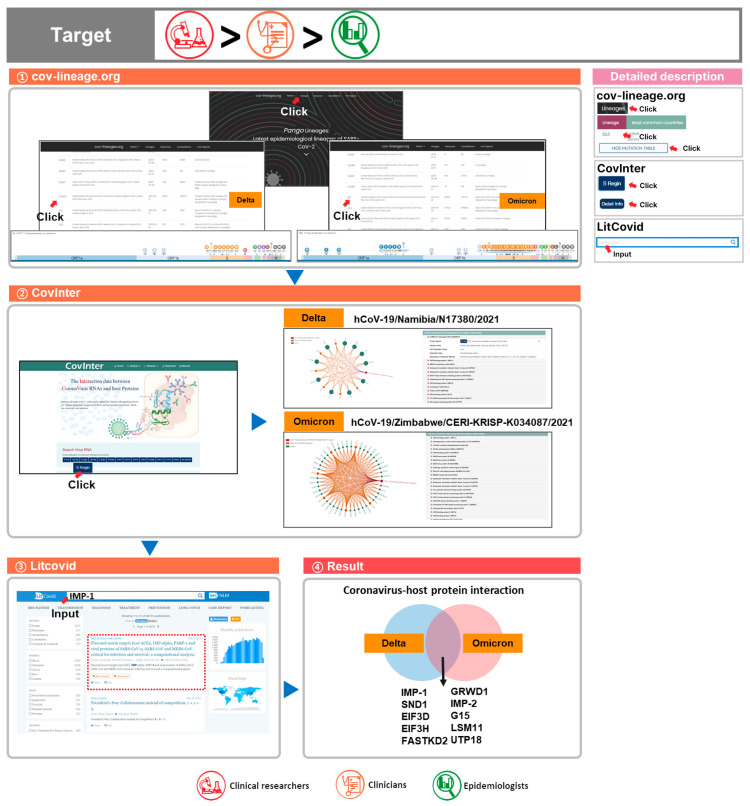
Comprehensive utilization of COVID-19 databases 2. To analyze the differences between the two lineages (Delta b.1.617.2 and Omicron BA.1) and find proteins that interact with the variants, the following process was carried out.

**Figure 4 viruses-15-00692-f004:**
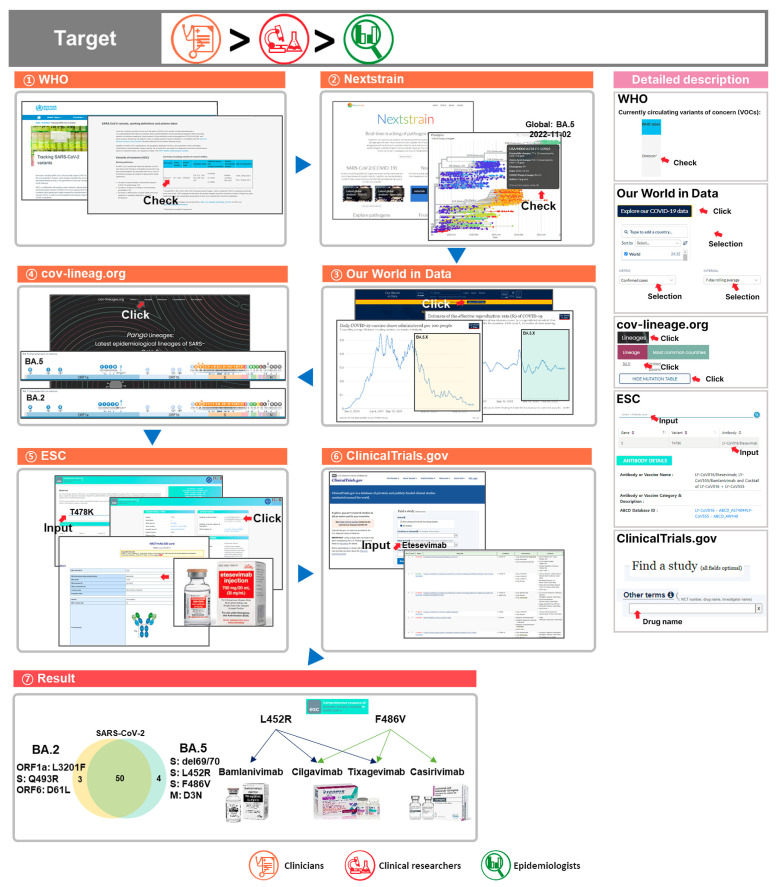
Comprehensive utilization of COVID-19 databases 3. To find data on antibodies and drugs that deal with variants, the following was carried out.

**Figure 5 viruses-15-00692-f005:**
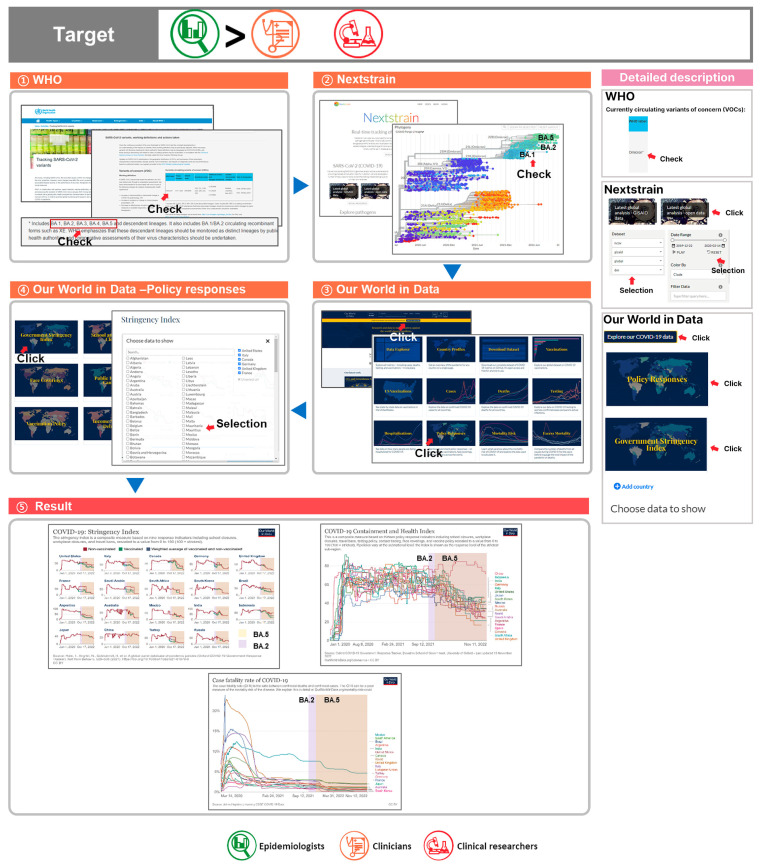
Comprehensive utilization of COVID-19 databases 4. Country policies and responses against SARS-CoV-2 variants were evaluated.

**Table 1 viruses-15-00692-t001:** Data volume categorized into genome and protein data, epidemiological data, and drug and target data.

Epidemiological data (global)		
World Health Organization	Cases: 623,893,894 Deaths: 6,553,936 Vaccinations: 12,814,704,622	18 October 2022
Johns Hopkins Coronavirus Resource Center	Cases: 627,632,333 Deaths: 6,578,449 Vaccinations: 12,821,432,441	24 October 2022
Our World in Data	Cases: 627.54 million Deaths: 6.58 million Vaccinations: 12.86 billion	23 October 2022
Genome and protein data		
GISAID (Global Initiative on Sharing Avian Influenza Data)	10 million genome sequences of SARS-CoV-2 submitted to EpiCoV	April 2022
	Audacity Global Phylogenetic tree comprised of 10,703,377 high quality genomes	26 October 2022
	CoVizu High-quality genomes: 7,726,056	24 October 2022
Nextstrain	Latest global SARS-CoV-2 analysis (GISAID data): 2943 genomes	27 October 2022 Collected December 2019–October 2022
	Latest global SARS-CoV-2 analysis (open data): 3006 genomes	27 October 2022 Collected Dec 2019–Oct 2022
	Phylogeny of SARS-like betacoronaviruses including novel coronavirus SARS-CoV-2: 49 genomes	27 October 2022
COVID-19 Data Portal	Viral sequences: Data types (14,849,714)	27 October 2022
	Host sequences (human and other hosts): Host sequences (30,694)	
	Expression: Data type (226)	
	Proteins (3772)	
	COVID-19 pathways, interactions, complexes, targets and compounds: Data types (7801)	
NIH	5,808,129	27 October 2022
	Source DNA (159,494) RNA (5,642,181)	
	Type Exome: 2 Genome: 65,958	
Treatment (Clinical trials, drug)		
NIH	ClinicalTrials.gov (8357 studies)	27 October 2022
	Status Completed: 3013	
	Study phase Early phase 1: 61 Phase 1: 671 Phase 2: 1494 Phase 3: 929 Phase 4: 258 Not applicable: 1920	
Literature		
COVID-19 Data Portal	Literature (805,970)	21 October 2022
NIH	PubMed: 307,261 results	27 October 2022
	PMC (PubMed Central): 429,092 results	
	LitCOVID: 300,174 publications in PubMed, 8000 journals	

**Table 2 viruses-15-00692-t002:** COVID-19 database addresses and associated publications.

No.	Website	Address (Database Utilization)	Ref
1	GISAID	https://gisaid.org/	[32]
2	Cov-lineages.org	https://cov-lineages.org/	[33]
		Pangolin: https://cov-lineages.org/resources/pangolin.html	
		Scorpio: https://github.com/cov-lineages/scorpio	
		Pando: http://pando.tools/	
		Civet: https://cov-lineages.org/resources/civet.html	
		Polecat: https://github.com/artic-network/polecat	
		pango.network: https://www.pango.network	
3	COVID CG	https://covidcg.org/	[34]
4	COVID-19 Data portal	https://www.covid19dataportal.org/	[35]
5	Nextstrain	https://nextstrain.org/	[36]
6	NCBI SARS-CoV-2 Resources	https://www.ncbi.nlm.nih.gov/sars-cov-2/	
		LitCOVID: https://www.ncbi.nlm.nih.gov/research/coronavirus/	[37]
		PubMed: https://pubmed.ncbi.nlm.nih.gov/?term=covid-19	
		PubMed Central: https://www.ncbi.nlm.nih.gov/pmc/about/covid-19/	[38]
		BLAST: https://blast.ncbi.nlm.nih.gov/Blast.cgi?PAGE_TYPE=BlastSearch&BLAST_SPEC=Betacoronavirus	
		PubChem: https://pubchemdocs.ncbi.nlm.nih.gov/covid-19	
		ClinicalTrials.gov: https://clinicaltrials.gov/ct2/home	
7	PDB	https://www.rcsb.org/	[39]
8	WHO	https://covid19.who.int/	[40]
9	COVID-19 Research at UCSC	https://genome.ucsc.edu/covid19.html	[41]
		Usher: https://github.com/yatisht/usher	[42]
		UCSC Genome Browser view of SARS-CoV-2 genomic datasets: https://genome.ucsc.edu/cgi-bin/hgTracks?db=wuhCor1&lastVirtModeType=default&lastVirtModeExtraState=&virtModeType=default&virtMode=0&nonVirtPosition=&position=NC_045512v2%3A1%2D29902&hgsid=1506921125_d8K9do0hsuR7zvE950cXSU3hQqYV	
		UCSC cell browser: https://genome.ucsc.edu/singlecell.html	[43]
10	Our World in Data	https://ourworldindata.org/	[44]
11	John Hopkins university coronavirus resource center	https://coronavirus.jhu.edu	[45]
12	DOCK CoV-2	https://covirus.cc/drugs/	[46]
13	T-CoV	https://t-cov.hse.ru	[47]
14	CovInter	http://covrpii.idrblab.net/	[48]
15	ESC	https://clingen.igib.res.in/esc/	[49]

## Data Availability

Not applicable.

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
