# Peer review of "Systematic Guidelines for Effective Utilization of COVID-19 Databases in Genomic, Epidemiologic, and Clinical Research"

_viruses, 2023, doi:10.3390/v15030692_

Round 1

Reviewer 1 Report

The article by Seong at al. is devoted to an important topic. I recommend this paper to be published in the journal. Here are some minor suggestions:

1: In line 42, it is suggested to add some background of this study in introduction and highlight the novelty of this work clearly. For example, “Effective measures, such as vaccines (Nature Immunol. 2022, 23, 360-370), small-molecule inhibitors (J. Med. Virol. 2022, 94, 1373-1390), and natural products (Viruses. 2021, 13, 305; Biomedicines. 2021, 9, 689) are greatly needed to reduce SARS-CoV-2 transmission. However, promising drugs still do not exist (JAMA. 2022, 327, 2380-2382; Front. Immunol. 2022, 13, 1015355). The underwhelming clinical performance of current treatments do not mean that their significance can be disregarded. In contrast, the fairly large number of scientific literatures provides important insights for clinical advancement.” This is critical to address in this manuscript, the authors should enrich this part in the revised version.

2: The “Conclusions” section of the manuscript, drawbacks and challenges of COVID-19 databases should be discussed.

Author Response

Authors’ Responses to the Reviewer #1’s Comments

Thank you for the kind remarks and suggestions. We added a paragraph in the introduction reflection the reviewer’s suggestion, which we think improves the general flow and background of the research. We also expanded our Conclusions section to better reflect the findings of our work.

  1. In line 42, it is suggested to add some background of this study in introduction and highlight the novelty of this work clearly. For example, “Effective measures, such as vaccines (Nature Immunol. 2022, 23, 360-370), small-molecule inhibitors (J. Med. Virol. 2022, 94, 1373-1390), and natural products (Viruses. 2021, 13, 305; Biomedicines. 2021, 9, 689) are greatly needed to reduce SARS-CoV-2 transmission. However, promising drugs still do not exist (JAMA. 2022, 327, 2380-2382; Front. Immunol. 2022, 13, 1015355). The underwhelming clinical performance of current treatments do not mean that their significance can be disregarded. In contrast, the fairly large number of scientific literatures provides important insights for clinical advancement.” This is critical to address in this manuscript, the authors should enrich this part in the revised version.

Response: We added relevant background information to the introduction with appropriate references [11] and [12] (page1 line 44 – page 2 line 50). We commented on how the mRNA vaccine and COVID-19 treatments were developed quickly and how it is important to have a comprehensive understanding of COVID-19 data being accumulated.

  1. The “Conclusions” section of the manuscript, drawbacks and challenges of COVID-19 databases should be discussed.

Response: In the Conclusions section, we added a discussion of the challenges of COVID-19 databases, specifically in regard to how they are specialized and how difficult it is to access the data one requires. We also expanded on how we built queries and supplementary materials to address these drawbacks.

Reviewer 2 Report

It is very clear that the authors have put a very substantial amout of work into this paper and the idea behind the publication, of making all the work done on COVID-19 more readily accessible, is to be commended. However, I found this paper very difficult to read and interpret and I would ask if the authors could simplify their work in order to make it more reader-friendly.

Author Response

Authors’ Responses to the Reviewer #2’s Comments

Thank you for the feedback. We have made extensive efforts to address the points raised and raise the overall readability and quality of the paper.

  1. I think this needs qualifying

Response: We have added specifics about the Ebola and Zika viruses for previous outbreaks (page 2 line 56).

  1. What is this?

Response: We have fixed the indicated error in Table 1.

  1. Please redo these tables - the word split is not good.

Response: We have corrected and revised Table 2, addressing the word split issue and overall formatting.

  1. This needs to be written in full somewhere before you start using the abbreviation.

Response: We have added the full term of the abbreviated VOC (page 8 line 199).

  1. Since these figures are almost unreadable I do not know what they add to your paper.

Response:

We have added “Detailed description” panels to Figure 2, 3, 4, and 5 that enlarge indicated sections from the figure. Overall, we have made the figures more presentable and present information with more clarity.

  1. A similar comment to the previous one

Response: We have made the same corrections to Figure 3 as we did to the other figures.

  1. As before

Response: We have made the same corrections to Figure 4 as we did to the other figures.

  1. This is not an acceptable way to present your bibliography. Please check the journal requirements.

Response: We have updated the formatting of the references section to match the journal requirements.

  1. It is very clear that the authors have put a very substantial amount of work into this paper and the idea behind the publication, of making all the work done on COVID-19 more readily accessible, is to be commended. However, I found this paper very difficult to read and interpret and I would ask if the authors could simplify their work in order to make it more reader-friendly.

Response: We have improved upon the tables and figures to make the paper more reader friendly. To simplify the work, we included additional panels to the figures for accessibility. We also expanded the Conclusions section to make our overall scope and goal of the research clearer. This paper had already received revisions from a language editing service prior to the reviewer comments. To further improve the flow of the paper, we had a Native English-speaking colleague go over our work for language improvement.

Reviewer 3 Report

In this work, the authors present systematic guidelines for effective utilization of COVID-19 databases in genomic, epidemiologic, and clinical research.   The manuscript is overall readable, but the topic has already been discussed in the literature.   The authors are strongly requested to specify how this work differs from other surveys in the literature.    In the work, the authors present 15 databases but the paper lacks of comparison among these ones. The authors are requested to make this comparison.   Furthermore, authors do not discuss the utilization of COVID-19 databases in genomic, epidemiologic, and clinical research. I suggest to insert a Discussion Section where discuss that.   Also, the table should be reorganized in order  to make reading less difficult.   Finally, in the conclusion section, the authors affirm that the methodology investigated and explained in their study is expected to be used by various researchers and be the basis for building infrastructure and resources to effectively deal with any future  epidemics that may occur.   I suggest to explain deeply how it can help the research to manage the future epidemics.

Author Response

Authors’ Responses to the Reviewer #3’s Comments

Thank you for the helpful and extensive feedback. We have made the improvements to the Conclusion section, restating key points and addressing the significance of the research and our findings. In addressing the comments provided by the reviewer, we have made enrichments to the paper.

  1. In this work, the authors present systematic guidelines for effective utilization of COVID-19 databases in genomic, epidemiologic, and clinical research. The manuscript is overall readable, but the topic has already been discussed in the literature. The authors are strongly requested to specify how this work differs from other surveys in the literature.

Response: Our review investigated and categorized pre-existing COVID-19 databases, illustrating their characteristics, and highlighting the specified and disparate nature of them. The queries we presented show how to utilize databases without special expertise. In the Conclusions section, we have included statements to stress these points once again.

  1. In the work, the authors present 15 databases but the paper lacks of comparison among these ones. The authors are requested to make this comparison.

Response: In our manuscript, we compared data amount, data type, and database purpose for the individual databases. Tables 1, 2, and Figure 1 collectively serve to illustrate the unique characteristics of the 15 databases and the differences between them. The Supplementary materials provide detailed explanations on navigating and utilizing each database. We have also added a statement in the Conclusions section.

  1. Furthermore, authors do not discuss the utilization of COVID-19 databases in genomic, epidemiologic, and clinical research. I suggest to insert a Discussion Section where discuss that.

Response:

In our manuscript, we illustrate the wide range of COVID-19 data available that has been created, and the challenges in organizing it. Our queries are presented as methods of linking and utilizing COVID-19 data with ease. In the Conclusions section, we reinstate how finding and suggesting necessary treatment options is facilitated by COVID-19 data.

  1. Also, the table should be reorganized in order to make reading less difficult.

Response: We have corrected and revised Table 2, making it more presentable and easier to read.

  1. Finally, in the conclusion section, the authors affirm that the methodology investigated and explained in their study is expected to be used by various researchers and be the basis for building infrastructure and resources to effectively deal with any future epidemics that may occur. I suggest to explain deeply how it can help the research to manage the future epidemics.

Response: Our research illustrates how organizing COVID-19 data can help the scientific community deal with COVID-19 in various ways. In our Conclusions section, we expanded upon how our methodology can help future epidemic research as well. With a new epidemic, epidemiological data and genomic data will prove to be important in understanding and combating new diseases.